# Heterogeneity of Sonic Hedgehog response dynamics and fate specification in single neural progenitors

**Fengzhu Xiong[1,2]\*[†], Andrea R Tentner[1†], Sandy Nandagopal[1], Tom W Hiscock[1,3], Peng Huang[4], Sean G Tsung-Megason[1]\***

[1]Department of Systems Biology, Harvard Medical School, Boston, United States; [2]Gurdon Institute and Department of Physiology, Development and Neuroscience, University of Cambridge, Cambridge, United Kingdom; [3]Institute of Medical Sciences, School of Medicine, Medical Sciences and Nutrition, University of Aberdeen, Aberdeen, United Kingdom; [4]Department of Biochemistry and Molecular Biology, University of Calgary, Calgary, Canada

\*For correspondence:
fx220@cam.ac.uk (FX);
megason@hms.harvard.edu
(SGT-M)

[†]These authors contributed
equally to this work

**Competing interest:** The authors
declare that no competing
interests exist.

Reviewing Editor: Giulia Boezio,
The Francis Crick Institute,
United Kingdom

## eLife Assessment

This study presents an **important** study of the relationship between morphogen signaling and cell fate choices in the forming zebrafish neural tube, addressing a topical question in developmental biology. The authors provide a **solid** characterization of the precision limit for gene regulatory networks interpreting Shh, with single-cell resolution and state-of-the-art in vivo approaches. While the depth of analysis is restricted, particularly by the number of cell traces, the study will be of interest to developmental biologists interested in cellular decision-making.

**Abstract** During neural tube patterning, a gradient of Sonic hedgehog (Shh) signaling specifies ventral progenitor fates. The cellular response to Shh is processed through a genetic regulatory network (GRN) to specify distinct fate decisions. This process integrates Shh response level, duration, and other inputs and is affected by noise in signaling and cell position. How reliably the Shh response profile predicts the fate choice of a single cell remains unclear. Here, we use live imaging to track neural progenitors in developing zebrafish and quantify both Shh and fate reporters in single cells over time. We found that there is significant heterogeneity between Shh response and fate choice in single cells. We quantitatively modeled reporter intensities to obtain single-cell response levels over time and systematically evaluated their correlation with cell fate specification. Motor neuron progenitors (pMNs) exhibit a high degree of variability in their Shh responses, which is particularly prominent in the posterior neural tube where the Shh response dynamics are similar to those of the more ventrally fated lateral floor plate cells (LFPs). Our results highlight the precision limit of morphogen-interpretation GRNs in small and dynamic target cell fields.

## Introduction

Morphogens pattern tissue by acting over space in a concentration-dependent manner (*Wolpert, 1969*; *Kicheva and Briscoe, 2023*). Biochemically, morphogen molecules may either regulate transcription directly (*Driever and Nüsslein-Volhard, 1989*; *Driever et al., 1989*) or through a specific transducer following a cascade of intracellular interactions – a signaling pathway. The latter involves additional, potentially nonlinear steps of signaling. A prominent example of morphogen patterning is the vertebrate ventral neural tube where the morphogen Sonic hedgehog (Shh), produced from the

notochord and floor plate, forms an extracellular gradient. Through its transcriptional effector Gli, this Shh gradient specifies different neural progenitor domains along the ventral-dorsal axis (*Jessell, 2000*, *Figure 1A*). In this system, Gli activity profiles (cellular responses) regulate the expression of multiple homeobox transcription factors (fate genes) (*Briscoe et al., 2000*). These fate genes regulate their own and each other's expression as a genetic regulatory network (GRN); this network finally settles into one of several discrete and stable states that correspond to different neuronal fates (*Lek et al., 2010*; *Balaskas et al., 2012*). On the basis of population-level observations of the Shh pathway and transcriptional network, different aspects of the Shh signaling/Gli activity profile have been proposed to control cell fate choices: e.g., threshold levels (presumably by different affinities between Gli and fate gene enhancers, *Ericson et al., 1997*; *Oosterveen et al., 2012*), duration of response (sustained Gli levels under the adaptive negative feedback of the Patched [Ptc] receptor, *Dessaud et al., 2007*), and the time-integrated response level (*Dessaud et al., 2010*). As these response dynamics are integrated through the GRN toward fate decisions, inherent noise in gene expression (*Rosenfeld et al., 2005*) may generate heterogeneity in the response-fate correlation, despite the ability of the GRN to increase robustness (*Balaskas et al., 2012*). In addition, our previous work reveals extensive cell movement (positional noise) and a role of cell sorting in refining the boundaries of fate domains in the zebrafish neural tube after cells are specified (*Xiong et al., 2013*), which are mediated by an adhesion code downstream of cell fate choices (*Tsai et al., 2020*). These studies suggest a possible limit of precision in morphogen signal interpretation in fate specification. To test this possibility, single-cell observations using live imaging of transgenic reporters to simultaneously monitor both morphogen response and cell fate choices in vivo are required (*Xiong and Megason, 2015*). The zebrafish neural tube, with established reporter lines of Shh signaling and neural progenitor fates (*Huang et al., 2012*; *Xiong et al., 2013*; *Tsai et al., 2020*), provides a feasible model system to perform this analysis.

## Results

### Live tracking of individual neural progenitor fate choices in zebrafish embryos

To enable a direct, single-cell quantification of Shh response dynamics and fate choice, we developed an imaging protocol using double transgenic reporters along with a nuclear or membrane localized EBFP2 for cell tracking (*Figure 1B*). We combined the *tgBAC(ptch2:kaede)* reporter whose Kaede expression reports Shh response (*Huang et al., 2012*) and a second transgenic reporter (e.g. *tg(nkx-2.2a:mgfp)* [*Ng et al., 2005*], *tg(olig2:gfp)* [*Shin et al., 2003*, *Figure 1C*]) whose expression levels indicate different fate choices (*Xiong et al., 2013*). Movies with good spatial-temporal coverage of the neural plate/tube (e.g. *Video 1*) were analyzed by single-cell tracking in GoFigure2, an open-source software we previously developed for image analysis (*Xiong et al., 2013*). By manually placing a seed at the center of the same cell at each time point in the dataset, we generated a series of spherical segmentations inside the cell, which capture the fluorescence intensity of each reporter and the position of the tracked cell over time. Together, this time-dependent positional and reporter intensity information represents what we call a 'neural progenitor track' (*Figure 1D and E*, *Video 2*). We analyzed over 200 randomly chosen tracked cells to cover the ventral progenitor types in the zebrafish neural tube: the medial floor plate cells (MFPs), lateral floor plate cells (LFPs), motor neuron progenitors (pMNs), and 'more dorsal' cells (e.g. p2, p1, p0 progenitors) (*Figure 1A*; *Xiong et al., 2013*). These cell fates form distinct domains highly conserved among vertebrates and prescribe the pattern of neurogenesis (*Jessell, 2000*; *Lewis and Eisen, 2003*). In zebrafish, MFPs along with the notochord are the source cells of Shh; LFPs, pMNs, and more dorsal cells depend on Shh signaling for correct specification (*Park et al., 2004*; *Huang et al., 2012*). Taking advantage of the reporter expression and the stereotypic position at the end time point of each track, we identified and assigned different fates to the tracked neural progenitors (*Xiong et al., 2013*). We imaged cells from both the anterior spinal cord (approximately somite levels 4–9) and the posterior spinal cord (approximately somite levels 9–14). The shape, size, and morphogenetic movements of both the Shh sending and receiving tissues vary along the AP axis (*Xiong et al., 2013*), thus allowing us to sample a diversity of input-output (i.e. Shh response-fate choice) relationships.

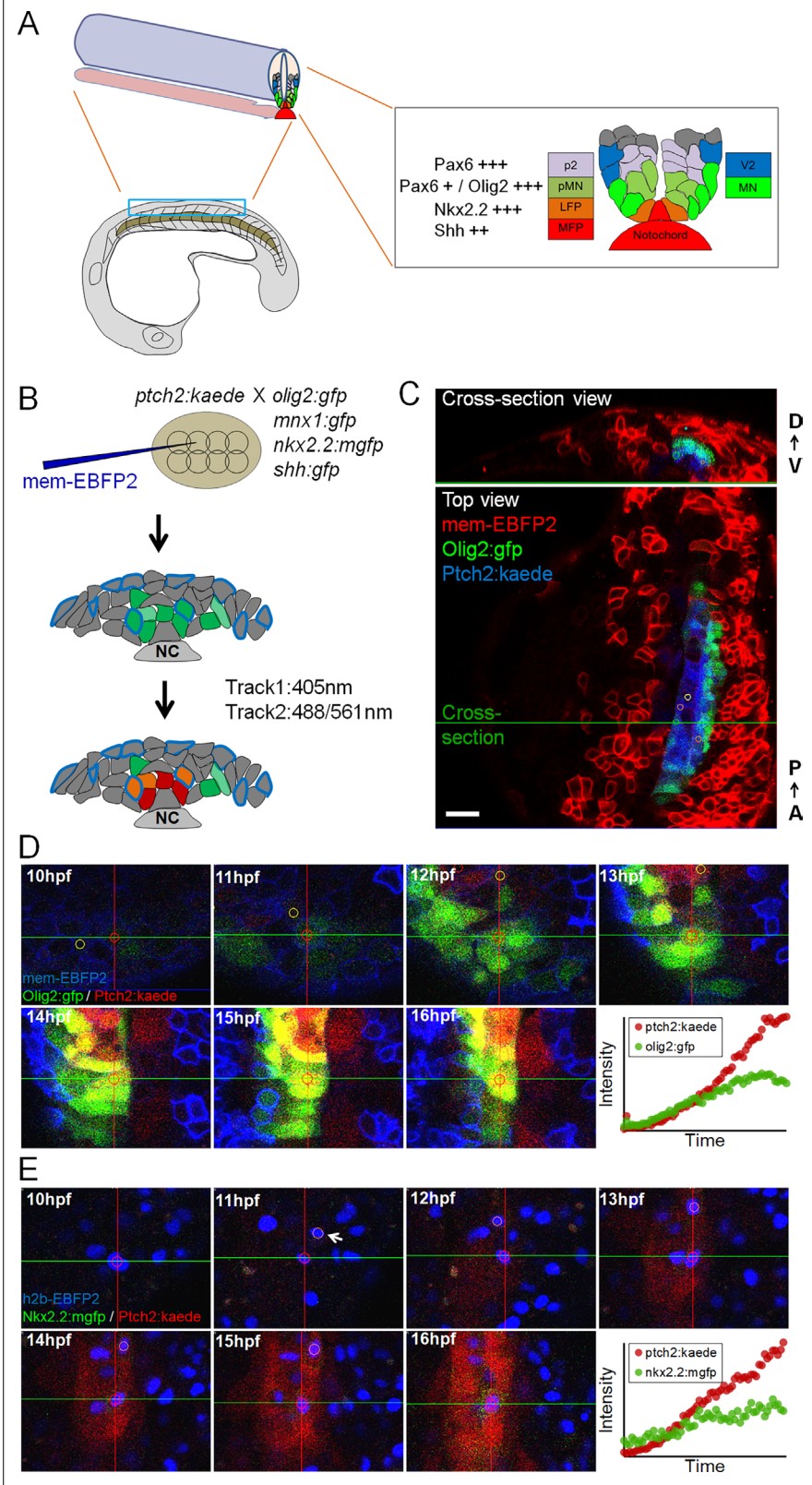

**Figure 1.** Imaging zebrafish neural tube development and single-cell tracks of different fates. (**A**) Schematic of ventral neural tube anatomy and pattern in zebrafish (partially adapted from *Xiong et al., 2013*). MFP, medial floor plate; LFP, lateral floor plate; pMNs, motor neuron progenitors (MN, motor neurons); p2, V2 interneuron progenitors (V2, V2 interneurons). (**B**) Labeling (at 8- to 16-cell stage) and imaging (at neural plate, neural keel, and

*Figure 1 continued on next page*

*Figure 1 continued*

neural tube stages) scheme for acquiring three channels at the same time. The 405 laser in the first scan line excites the BFP cell tracking marker, as well as converting Kaede (green) to Kaede (red). NC, notochord. (**C**) Example data. Small circles on the image show manual segmentations generated on cells using GoFigure2 software. D-V, dorsal-ventral; A-P, anterior-posterior. Scale bar: 20 μm. (**D**) A pMN track in a mem-ebfp2, *ptch2:kaede/olig2:gfp* dataset. Circles in the cells show a 2D view of the spherical segmentations generated using GoFigure2 software. (**E**) An LFP track in a h2b-ebfp2, *ptch2:kaede/nkx2.2:mgfp* dataset. Arrow indicates a sister cell of the centered track. Note that while this transgenic reporter has a membrane tag, the spherical segmentation is still capable of measuring the much weaker cytoplasmic fluorescent signal. See also *Video 1*.

## Heterogeneity is pervasive in Shh response dynamics and fate choice in single cells

Our set of ~200 single-cell tracks provides a first glimpse into the dynamics of Shh interpretation in vivo. The main new insights come from being able to observe the dynamics of two nodes of the GRN in the same live cell, namely a Shh response reporter (*ptch2:kaede*) and a fate reporter. To compare our data with previous studies, we first analyzed the average Shh response behavior of cells within each fate group (calculated by simple averaging of single-cell tracks within each of the different fates). In both the anterior and posterior spinal cord in zebrafish (*Videos 3 and 4*, respectively), the average LFP shows the fastest increase of Kaede intensity, while the average pMN shows intermediate increase and the average more dorsal cell is the slowest (*Figure 2A and C*). Accordingly, the *olig2:gfp* reporter is highly expressed in the average pMN but comes to a plateau after initial expression in the average LFP (*Figure 2B and D*). These data are consistent with observations that these cell types require different levels of Shh for specification (*Ericson et al., 1997*; *Park et al., 2004*) and that in amniotes the p3 progenitors (equivalent to LFPs in fish) initially express the pMN marker *olig2* but then shift to express *nkx2.2* with continued and/or increasing Shh exposure (*Jeong and McMahon, 2005*; *Dessaud et al., 2007*).

We next ask whether single cells behave stereotypically like the averaged cell. Strikingly, at the single-cell level, we observed high variability in Shh signaling reporter dynamics across cells of the same fate and some overlap of cells that finally take on different fates (*Figure 2A'–D'*). There are examples of cells that have similar *ptch2:kaede* traces but distinct fate reporter traces (e.g. in *Figure 2E*, track 1 vs 2, 3, vs 4), as well as cells with very different *ptch2:kaede* traces but similar fate reporter traces (e.g. in *Figure 2E*, track 1 vs 3). These data show that single cells have

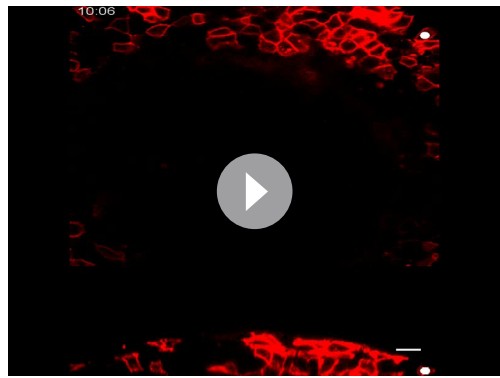

**Video 2.** Slice movie showing a single motor neuron progenitor (pMN) track in a mem-ebfp2, *ptch2:kaede/mnx1:gfp* dataset. Using a track as input, the movie was automatically generated by re-slicing the dataset centering on the tracked cells (marked by the white dot). The slices thereby allow visual observation of the cell's behavior even as it changes position. This cell is seen to move toward the midline and turn on the GFP reporter, adopting a pMN fate. Top view and cross-sectional view. Time stamp: hh:mm. Scale bar: 10 μm. Same below.

https://elifesciences.org/articles/96980/figures#video2

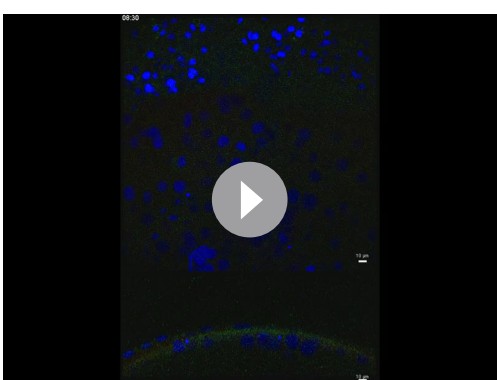

**Video 1.** h2b-ebfp2, *ptch2:kaede/nkx2.2:mgfp* dataset. 3D rendered maximum projections. Top view and cross-sectional view. Nuclear EBFP2 label allows easier tracking and identification of *nkx2.2:mgfp* cells.

https://elifesciences.org/articles/96980/figures#video1

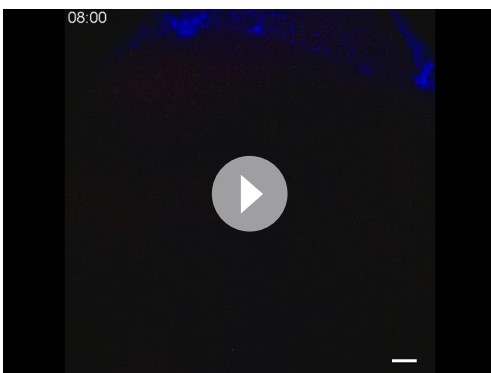

**Video 3.** mem-ebfp2, *ptch2:kaede/olig2:gfp* anterior dataset. Anterior neural tube formation. Epiboly movement and convergence are apparent in the beginning of the movie.
https://elifesciences.org/articles/96980/figures#video3

heterogeneity in the correlation between Shh response and fate choice. Cells that choose the same fate could have distinct Shh responses, and cells that have very similar Shh responses may make different fate choices.

## Quantifying reporter activity dynamics from fluorescent intensity tracks

Given the apparent heterogeneity and overlap in the single-cell track data, are there certain features in the reporter dynamics that predict fate outcome more accurately? In other words, are the apparently very similar *ptch2:kaede* tracks (as seen in *Figure 2E*) different in a certain way that could explain their distinct fate choices? Our tracks offer us an opportunity to address this question by quantifying and systematically classifying Shh signaling activity, namely the transcription rate of the *ptch2* reporter, from the fluorescence intensities. However, the intensity measurements must be handled carefully and interpreted with consideration of experimental limitations to provide useful insights.

First, while the fluorescence intensity levels measured in single cells by image segmentation are in principle proportional to the reporter protein concentration (*Figure 3A*), the acquisition and analysis processes may introduce several errors: bleaching, bleed-through, saturation, and segmentation measurement error. Our sampling frequency is even lower here than our previous work, in which other imaging settings were similar and quantification of fluorescence signals indicated that bleaching was not significant (*Xiong et al., 2013*), suggesting that the bleaching factor can be ignored here. To account for bleed-through, which refers to the phenomenon of signal from one fluorophore contributing to another during simultaneous acquisition, we applied a percentage subtraction from each channel after determining the bleed-through ratio using all data points (Materials and methods). To account for saturation, which refers to signal intensity exceeding the dynamic range of detection causing an artificial signal plateau, we removed data points where the fraction of saturated pixels exceeds 5% within segmented cells (Materials and methods). To handle noise that arises from random fluctuations of segmentation measurement between time points (technical errors), we applied a moving-average smoothing to the tracks over windows of 6 time points (36 min, see Figure 7C for effects of varying this time window). This operation causes some short-time scale features of the tracks (which are otherwise not possible to analyze due to technical errors in the measurements) to be lost, but it preserves trends on longer time intervals (*Figure 3B*). We further confirmed the mitigation of the impact of technical errors by re-tracking the same cells, where the average variation of intensity measurement over 6 time points is consistently <5% (data not shown). Together, this pipeline transforms raw tracks to an intensity track that reasonably approximates the reporter protein concentration dynamics in single cells. Note that other factors (e.g. cell growth, elongated cell shapes) may still affect the fluorescence intensity measurement independently of Shh signaling activity and not accounted for with these datasets that do not contain a cell membrane label.

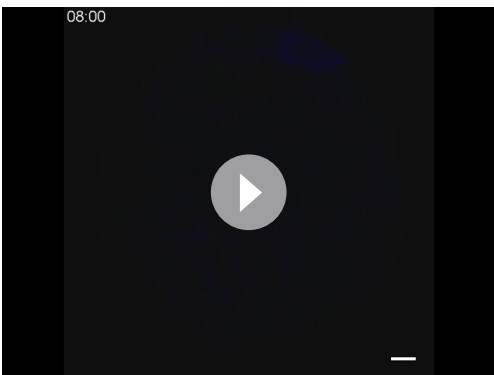

**Video 4.** mem-ebfp2, *ptch2:kaede/olig2:gfp* posterior dataset. Posterior neural tube formation. Blastopore closure is apparent in the beginning of the movie. Tissue is overall smaller than the anterior counterpart (*Video 3*).
https://elifesciences.org/articles/96980/figures#video4

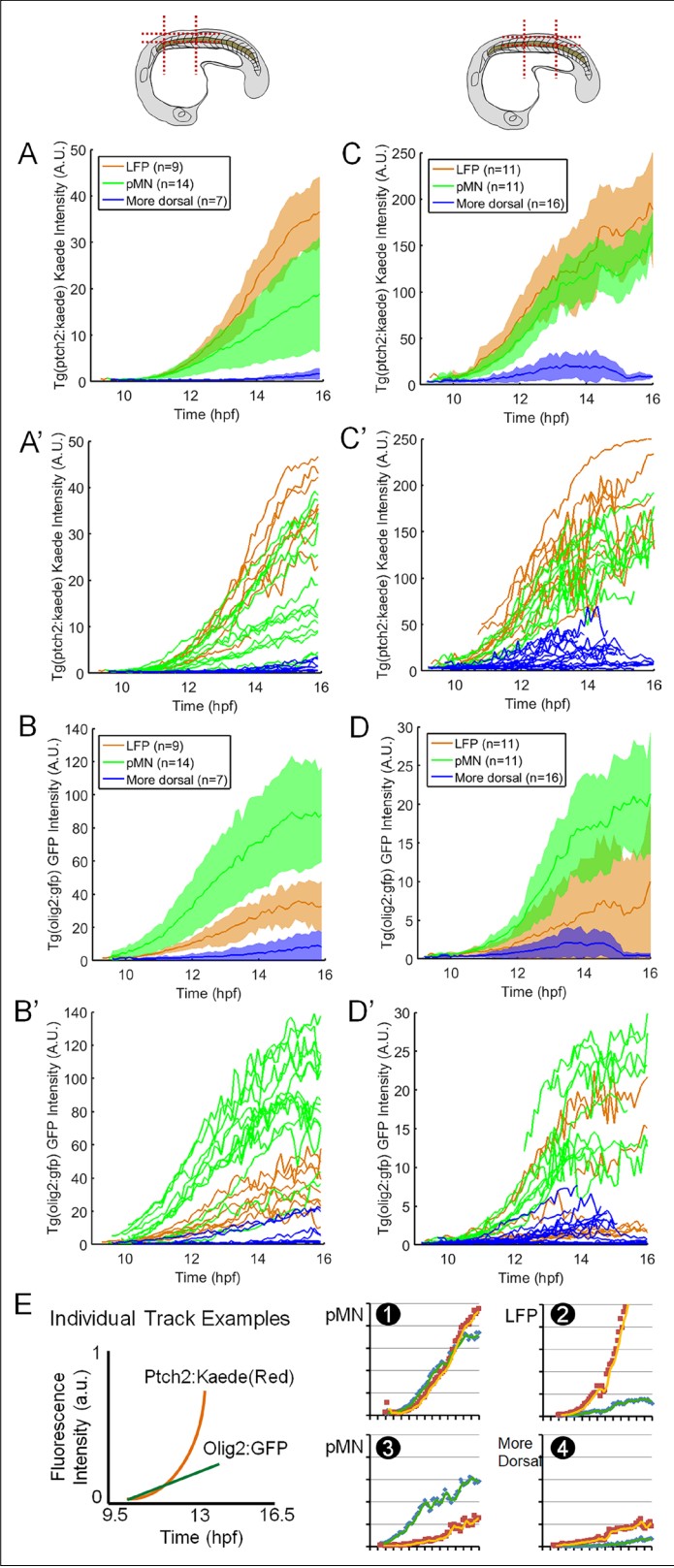

**Figure 2.** Dynamics and heterogeneity of Sonic hedgehog (Shh) response and fate reporters in average cells and single cells. (**A–D**) Average and single (**A'–D'**) tracks from a *ptch2:kaede/olig2:gfp* movie focusing on the anterior spinal cord (indicated by the schematic, **A–B**), and the posterior spinal cord (**C–D**), respectively. Colored shades around the average tracks represent ± SD. Individual *ptch2:kaede* traces (**A', C'**) show significant overlap between

*Figure 2 continued on next page*

*Figure 2 continued*

fates. Note that intensity units are not comparable between different datasets (e.g. between **A** and **C**) due to variation in expression, sample depth, mounting position, and imaging settings. The average tracks appear noisier before 10 hpf and after 15 hpf due to some single-cell tracks not being long enough temporally (e.g. a tracked cell leaves the field of view); therefore, fewer data points are available for averaging at the ends. LFPs, lateral floor plate cells; pMNs, motor neuron progenitors. See also *Videos 3 and 4*. (**E**) Example single tracks that show very similar *ptch2:kaede* response dynamics but very different *olig2:gfp* expression, and vice versa. The schematic shows the axis ranges and labels, and legends for the two reporter traces.

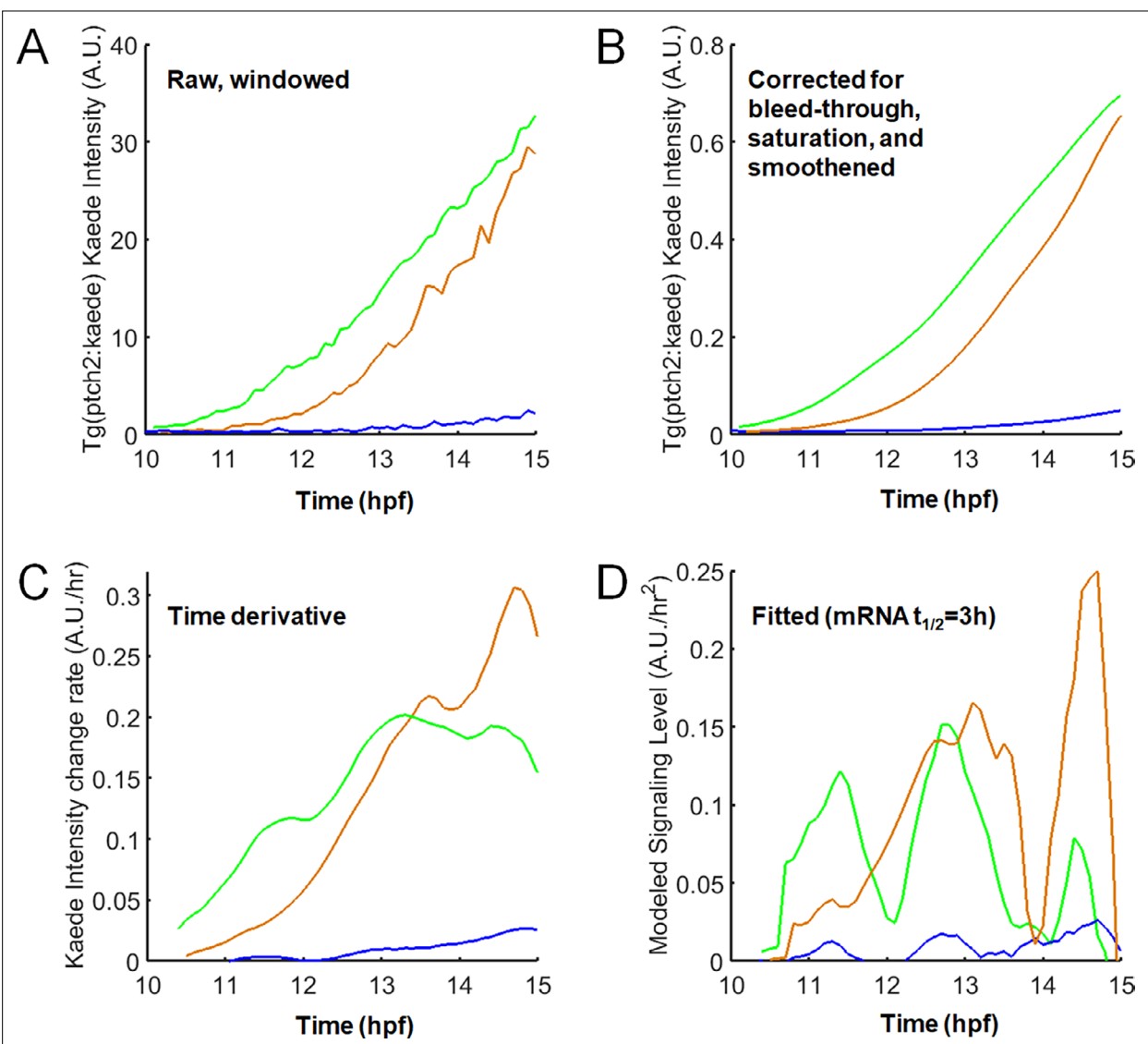

**Figure 3.** Estimation of dynamic reporter activity from fluorescence intensity measurements. (**A**) Example raw tracks (a motor neuron progenitors [pMN] [green], a lateral floor plate cell [LFP] [brown], and a more dorsal cell [blue] each). The tracks were trimmed to contain only data points between 10 and 15 hpf as this time window is best covered in the tracks. Before 10 hpf, the signal is low and by 15 hpf, most progenitors have become specified (*Xiong et al., 2013*). (**B**) Tracks in (**A**) after multiple filter corrections as described in the text. Main trends over long time windows (>0.5 hr) are preserved while short fluctuations are removed by a moving-average smoothing operation. (**C**) Rate of change in intensity by the first time derivative of (**B**). (**D**) Modeled signaling level (*ptch2:kaede* transcription rate) of the same tracks using *Equation 3*. Note the different peak height and duration of different cells. Varying the half-life value between 1 and 5 hr does not substantially change these results (data not shown).

Second, in order to estimate the cellular Shh response/Gli activity dynamics (reflected by the transcription dynamics at the *ptch2:kaede* reporter), we investigated the relationship between fluorescent intensity (*I*) and the transcription rate of *kaede* (*x*) (*Elowitz and Leibler, 2000*). Since Kaede and GFP proteins can be considered as stable on the time scale of our experiments (*Ando et al., 2002*; data not shown), assuming a constant coefficient ($c_1$) for the rate of Kaede translation and maturation as a function of *kaede* mRNA level (*m*), the intensity change rate of Kaede (*Figure 3C*) is given by:

$$\frac{dI}{dt} = c_1 m \tag{1}$$

*m* is in turn a function of transcription and mRNA stability, given in the form of half-life ($t_{1/2}$):

$$\frac{dm}{dt} = x - m\frac{ln2}{t_{1/2}} \tag{2}$$

Combining *Equation 1* and *Equation 2*, we have a simplified relationship between measured fluorescence intensity and Shh response (represented by *x*):

$$cx = \frac{d^2 I}{dt^2} + \frac{dI}{dt} \cdot \frac{ln2}{t_{1/2}} \tag{3}$$

where *c* is a constant. Note that the fluorescent signal is time delayed due to lags caused by transcription, translation, and maturation of the fluorescent protein; therefore, our raw tracks do not show reporter responses (transcription) in real time. Importantly, use of destabilized RNA or proteins would not reduce this lag, but only diminish reporter intensity, thus making their measurement noisier. Furthermore, because of the time and detection sensitivity differences between Kaede and GFP, we are prevented from cross-correlating the two reporter channels on the same time axis. Despite these limitations, *Equation 3* still allows us to estimate the Shh response dynamics in each single cell and compare that between different cell fates within each dataset.

To validate *Equation 3* and estimate reporter mRNA half-life ($t_{1/2}$) in vivo, we first applied a heat-shock pulse on transgenic embryos with a *hsp70l:EGFP* transgene (*Figure 4A*; *Halloran et al., 2000*, constructed with an SV40 polyadenylation sequence like the *ptch2:kaede* transgene). This causes a defined pulse of transcriptional output at the reporter (*Rieger et al., 2005*). According to *Equation 3*, the GFP intensity change that follows should reflect the exponential decay of the *gfp* mRNA. Indeed, single-cell tracks we generated provide an excellent fit with the model's prediction (*Figure 4B*), allowing us to estimate the mRNA half-life to be around 1.7 hr. Next, we performed in situ hybridization of *kaede* after sequential chemical inhibition of Shh response using cyclopamine (thereby stopping new *kaede* mRNA synthesis) and compared the overall mRNA signal with controls for signal level over a series of time points (*Figure 4C*; *Huang et al., 2012*). This method suggests an ~3 hr mRNA half-life. The two estimations are thus congruent on an in vivo half-life of fluorescent reporter mRNAs somewhere between 1 and 4 hr (effects of varying this parameter are assessed in Figure 7B). These data support the validity of *Equation 3* and enable us to use it and the estimated half-life to infer the dynamics of the *ptch2* transcription rate (i.e. Shh signaling response) in our single neural progenitor tracks (*Figure 3D*). Note that as *Equation 3* contains the second derivative with respect to time of the intensity, this fitting is prone to measurement noise and requires good time resolution. However, due to our smoothing operation, sharp and short-time response changes have been filtered out. Our subsequent comparisons only consider those features that span over at least 0.5 hr windows (e.g. the peaks and valleys seen in *Figure 3D*), which are comprised of at least 5 time point measurements and describe trends that can readily be seen in the raw intensity track (*Figure 3B*). This compromise minimizes the impact of technical errors in the analysis at the expense of model resolution.

## Systematic analysis of Shh response heterogeneity and association with fate outcome

The processed tracks allow us to gain deeper insights into the characteristics and sources of heterogeneity. First, we performed K-means clustering of *ptch2:kaede* dynamics in all cells (*Figure 5A*). The resultant clusters are then compared in terms of their contribution to the LFP, pMN, and more dorsal fates (*Figure 5B*). We found that large clusters all show fate heterogeneity between 'adjacent' fates (LFP and pMN, pMN and more dorsal), such as Clusters 1, 2, 3, 5, and 7. In contrast, a given cluster

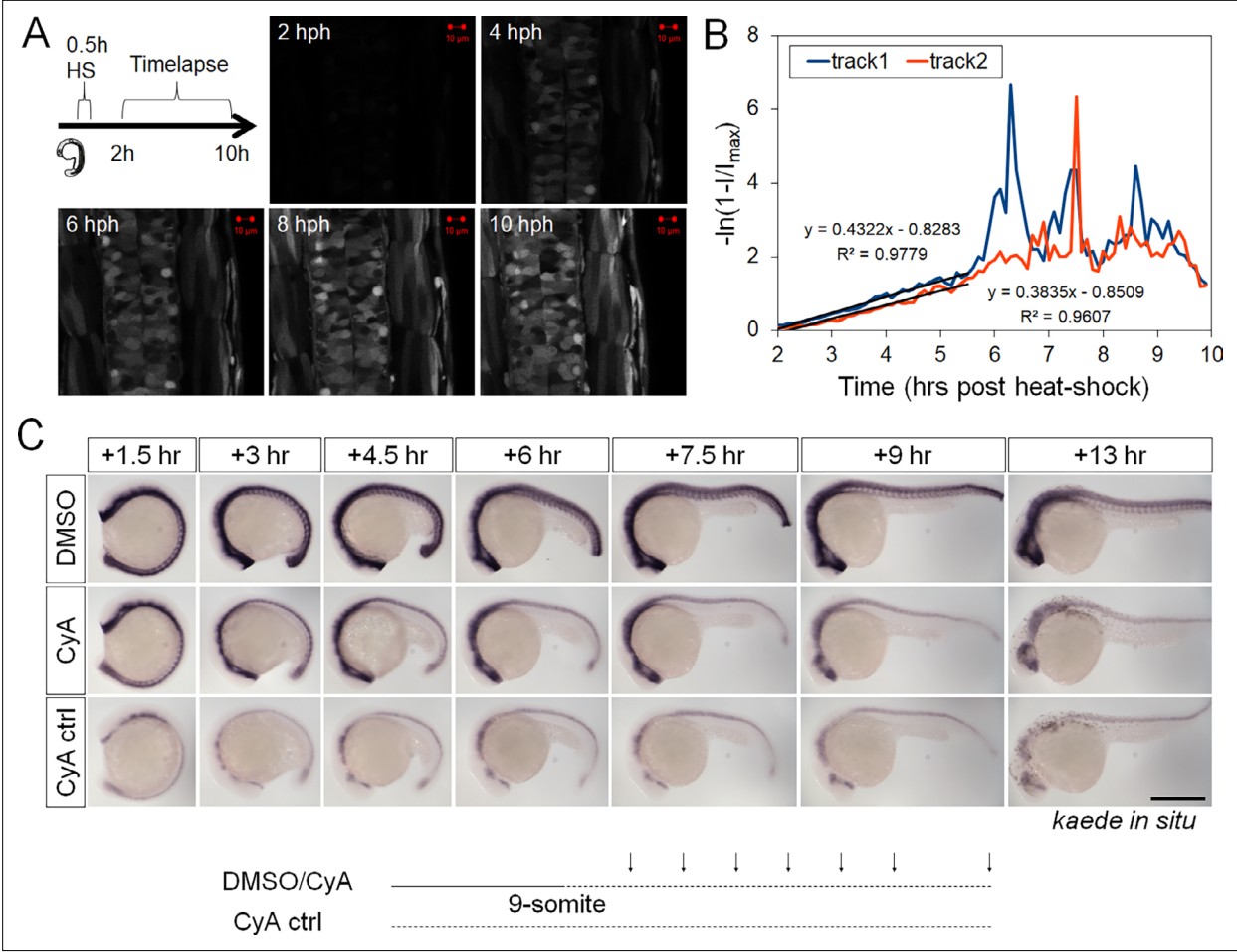

**Figure 4.** Estimation of fluorescent reporter mRNA half-life. (**A**) Heat-shock pulse induction of GFP and timelapse imaging (schematic). 24 hpf *tg(hsp70l:EGFP)* embryos were shocked for 30 min in pre-warmed 37°C egg water in a heat block. Imaging began 2 hr post heat-shock (hph, when the fluorescence was first detectable). (**B**) Cell tracks of hsp70l:EGFP to validate *Equation 3* and estimate mRNA stability. Assuming that the heat-shock between $t=0$ and $t=0.5$ causes a transient pulse of mRNA production, the intensity track then covers the simple decay period of mRNA (as no more mRNA is produced after the pulse). The decay is exponential according to *Equation 3* and governed by the mRNA half-life. The fluorescent intensity increase rate should then correspondingly decrease exponentially toward a plateau value, predicting a linear relationship between time and $-ln(1-I/I_{max})$, in which $I$ is raw intensity and $I_{max}$ is the plateau value of $I$ in later times (when no more mRNA is left and new fluorescent protein production stops). This predicted linear relationship is found for the tracks, suggesting the simple model correctly describes the process. The slope of the lines (~0.4) suggests the mRNA half-life of *hsp70l:egfp* to be ~1.7 hr. As $I$ approaches $I_{max}$, the quantity of $ln(1-I/I_{max})$ becomes sensitive to small fluctuations in $I$, resulting in the noisy spikes seen after 6 hr. This portion of the data was not used in the fitting. (**C**) Estimation of Kaede mRNA stability using in situ hybridization. CyA, cyclopamine, an antagonist of Shh signaling that represses ptch2 promoter output. When soaked in CyA (CyA ctrl), the embryos exhibit a low basal level of Kaede expression in the neural tube. When CyA was added around the 9-somite stage and embryos followed over time, the high level of Kaede mRNA as seen in the DMSO control was not maintained but decayed toward the basal level. In the trunk/tail neural tube (but not in the head), the level became similar to basal at +7.5 hr by comparing the CyA and CyA ctrl images. This result suggests Kaede mRNA is around for more than 6 hr after inhibition of signaling, with a half-life of about 3 hr. Scale bar: 400 μm.

does not contribute both to LFP and more dorsal, with the only exception in Cluster 11. These data show that the heterogeneity between Shh responses and fate choices is pervasive across distinct types (i.e. clusters) of response profiles, and primarily presents itself around progenitor domain boundaries. Over a larger spatial scale (such as between LFP and more dorsal), the response-fate relationship can be considered predictive and accurate at single-cell level.

To explore the sources of signal heterogeneity, we compared the average and variation of Shh response by fates in different datasets (*Figure 5C*). The pMN tracks showed the most variability in all datasets, while LFP and more dorsal tracks are more similar on average and less variable. A striking feature that stands out is the similarity between pMN and LFP tracks in the posterior dataset, where the average pMN response appears as strong as the LFPs. To account for the heterogeneity in timing

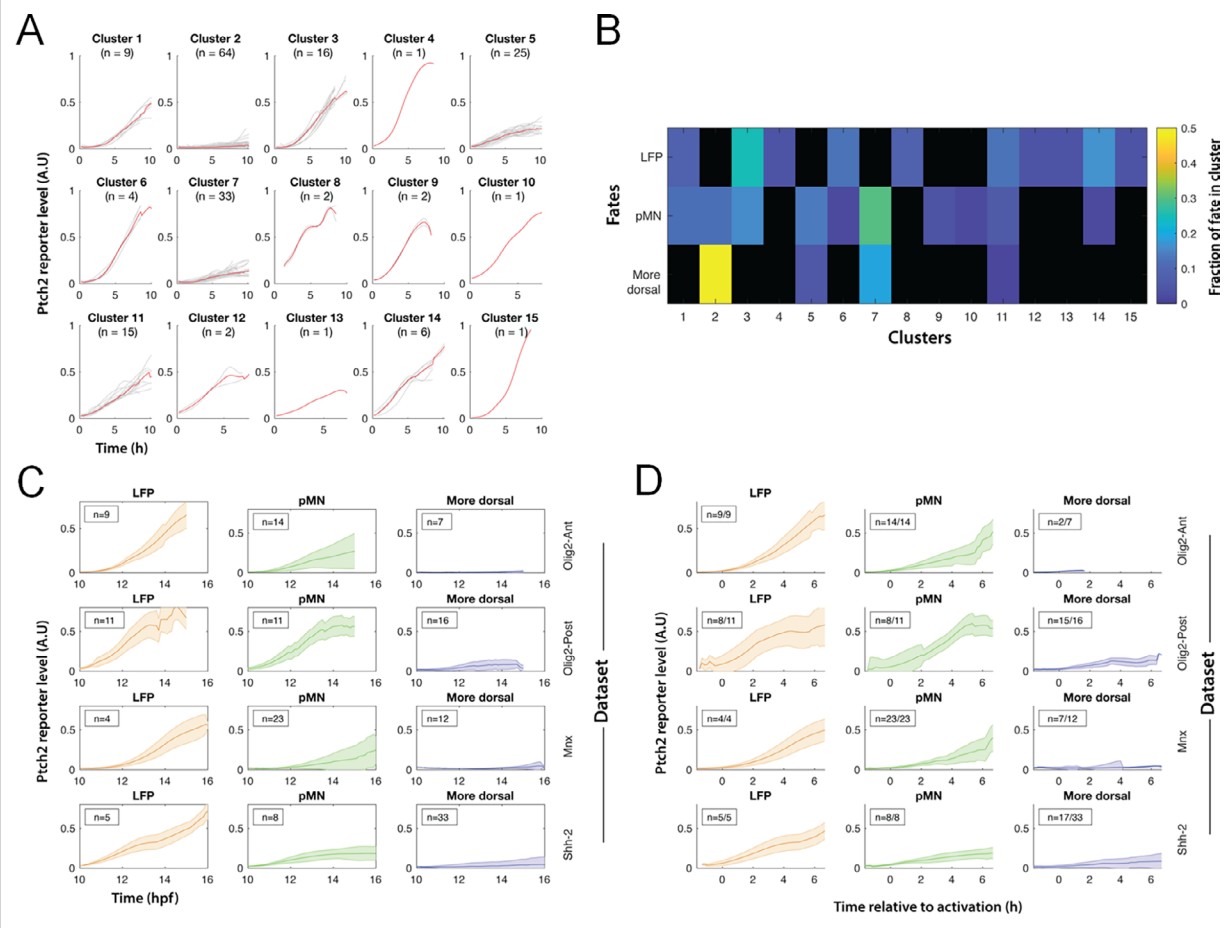

**Figure 5.** K-means clustering and temporal re-alignment analysis of tracks across datasets. (**A**) K-means clustering was performed on smoothed tracks (as in *Figure 3B*) from four datasets combining ptch2:kaede with different reporters (olig2:gfp, imaged anteriorly: 'Olig2-Ant' or posteriorly: 'Olig2-Post'; mnx1:gfp ('Mnx'); shh:gfp ('Shh-2')). (**B**) Fate contribution of clusters in (**A**). Heatmap shows, for each fate, the fraction of cells in different clusters. Black indicates no cells. (**C**) Average ptch2:kaede dynamics by cell fate and dataset from processed tracks. (**D**) Similar to (**C**), re-aligned starting times by the first time the modeled ptch2 promoter output exceeds a threshold of 0.025 of the normalized activity.

of response initiation, we used model-fit response levels (*Figure 3D*) to identify the response initiation time and re-aligned the tracks by this time point (*Figure 5D*). The re-aligned pMN average tracks are more similar to each other in the anterior datasets and show reduced variability, suggesting that some of the variability can be attributed to differences in timing of reporter activation. The posterior pMN tracks remain distinctly heterogeneous, suggesting a major contribution of variability is associated with the anterior-posterior (AP) level of the neural tube, which carries differences in tissue size and shape, cell movement, and morphogen source, among others.

## Correlation of key aspects of Shh response with fate outcome

To further compare the correlation of some key dynamic features of morphogen response to cell fate outcome between the anterior and posterior neural tube, we used the model-fit responses to calculate the maximum transient response level (the peak Shh response within the track), the average response level (average Shh response over 10–15 hpf), and the response time (the amount of time with an above-basal Shh response, which may include multiple durations of response) in the tracked time windows. To give a visual indication of how well each metric correlates with the graded fate outcome, we ranked single-cell tracks by the value of each metric high to low to make a vertical 'French flag': the sharper the flag, the stronger the correlation. The sharpness is quantified by the percentage of correct fate prediction by the metric thresholds. A value near 50% (coin toss) would suggest the metric does not correlate with fate choice at all, whereas 100% would suggest the metric

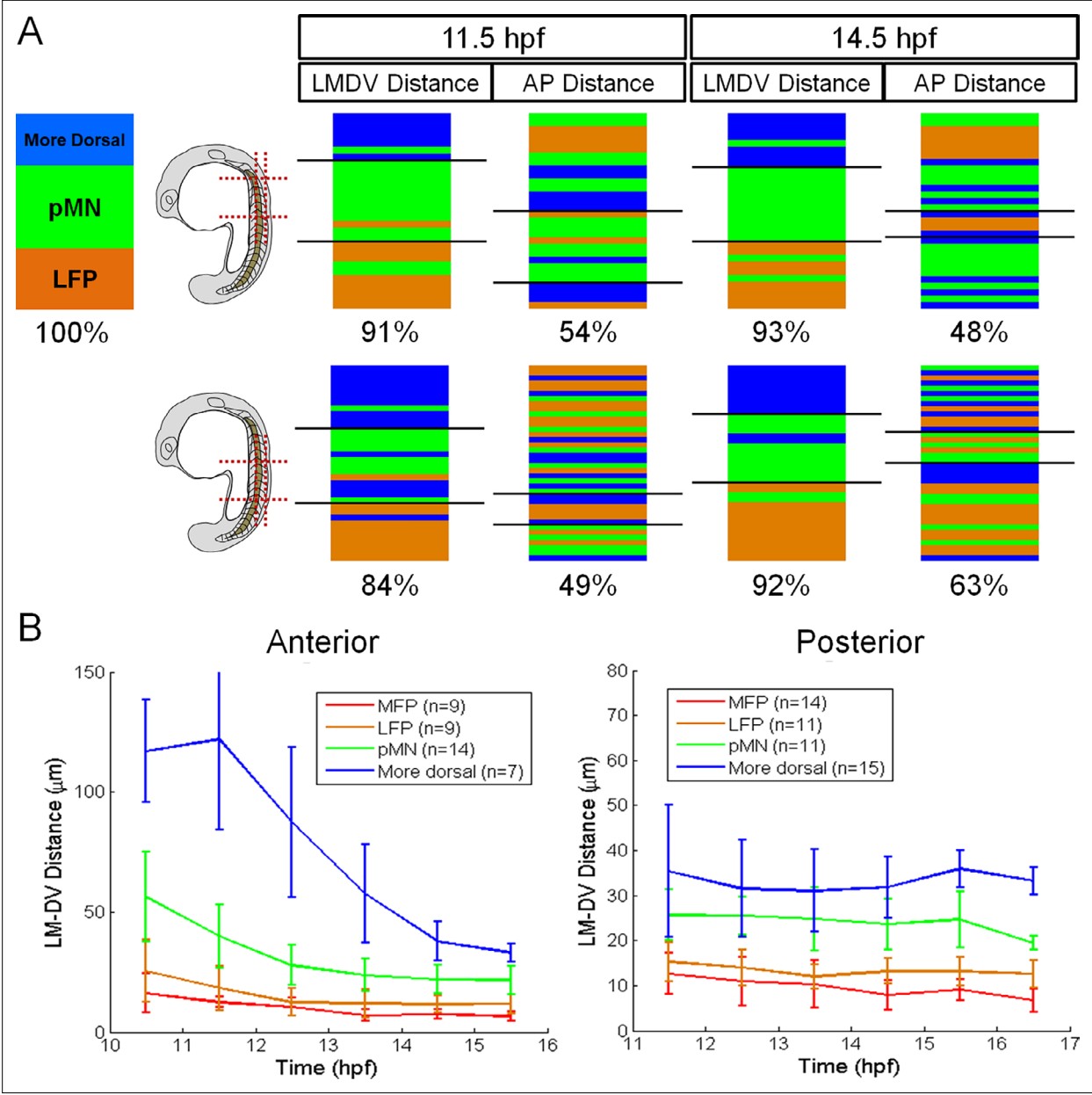

**Figure 6.** Correlations between position and fate choice in single cells. (**A**) Ranking of lateral-medial/dorsal-ventral (LMDV) and anterior-posterior (AP) positions as a control for the correlational analysis. Data from an anterior movie and a posterior movie are compared (schematics). The black lines on the flag mark thresholds that best separate different fates. See descriptions in the text and also Materials and methods. (**B**) Average position tracks. Note the range of anterior progenitors vs posterior ones over the same time window, indicating different tissue geometry and cell dynamics along the body axis. LMDV: lateral medial-dorsal ventral distances. Error bars are SD.

makes perfect prediction. For example, when we use two late-stage cell position metrics, the lateral-medial/dorsal-ventral (LMDV, this distance refers to the axis along which cells are being patterned which is initially lateral-medial because patterning begins at the neural plate which then gradually transitions to dorsal-ventral as the neural tube is formed) should do well in the flag correlation while the AP position should do poorly since we are trying to predict fates across the DV axis (*Figure 6A*). Indeed, we find that the LMDV position is highly correlated with cell fates. For the anterior tracks, the LMDV distance is consistently >90% correlated, while for the posterior tracks, the LMDV distance is more poorly correlated at earlier stages but becomes better at later times, consistent with a necessary role for cell sorting in sharpening fate domain boundaries over time (*Xiong et al., 2013*; *Tsai et al., 2020*). In contrast and as expected, the AP distances never correlate with fate choices in the anterior

or posterior spinal cord (all around 50%). The diversity of tracks and tissue geometry both within and across the anterior and posterior regions of the spinal cord (*Figure 6B*) enables us to search for a common metric that best correlates the Shh response dynamics with fate choice.

We compared the same sets of cell tracks by the maximum transient level, total time, and average levels of Shh response (*Figure 7A*), when ventral neural progenitors are known to become specified (*Lewis and Eisen, 2003*; *Park et al., 2004*; *Xiong et al., 2013*). For the anterior tracks, all metrics are >85% correlative with fate choices, consistent with previous population level studies in amniotes (*Ericson et al., 1997*; *Dessaud et al., 2007*; *Dessaud et al., 2010*). Interestingly, for the posterior tracks, only maximum transient level correlates with fate choice for >80% of the tracks, and the other two metrics fail to make 70%. This difference of correlation percentage is robust to wide variations in values of parameters used in our quantification (e.g. estimated Kaede mRNA half-life, smoothing parameters used in intensity calculation and transcription activity fitting) (*Figure 7B and C*). Other metrics (except LMDV cell position at late times, which is one of our fate identifiers) do not correlate better than the maximum transient level when both anterior and posterior tracks are considered (*Figure 7D*, data not shown). No single metric makes perfect predictions.

## Discussion

The imaging, cell tracking, and data analysis reported in this study open a fresh perspective for looking at the important classic problem of morphogen interpretation. While it is reassuring that our data (obtained with a distinct technique) on average corroborate numerous previous studies on Shh-mediated ventral neural tube patterning, the increased resolution into single cells and wide temporal coverage from when cells start responding to Shh to cell fate specification reveal a high degree of previously unrecognized heterogeneity. This single-cell variability could be due to other factors that differ between cells and their cross-talk with the Shh GRN, such as Notch signaling (*Huang et al., 2012*), or just simply noise that arises in the GRN (*Rosenfeld et al., 2005*). These possibilities are difficult to distinguish with the existing data, but they pose a challenge to morphogen patterning in general: how to ensure pattern precision despite this high heterogeneity in signal responses?

One possibility is to code morphogen response in a robust way under the given heterogeneity. Through quantifying the transcriptional dynamics at the Shh reporter, we are able to directly assess the correlation between certain proposed information-coding schemes and fate outcome. In the anterior spinal cord, maximum transient level, response time, and average level are all predictive of fates. In the more posterior spinal cord, however, our results show that the temporal dimension of Shh response (which is part of the response time and the average level) no longer explains cell fate choices, as well as the maximum transient level. The cause of this difference might be that there are intrinsic or environmental differences between neural progenitors in the anterior vs posterior spinal cord, which affect their morphogen interpretation. Supporting this idea, the tissue geometry and the range and movement of cells in zebrafish neural tube vary greatly along the body axis (*Xiong et al., 2013*; this study). We also found that some of the posterior LFP and pMN tracks still show very similar Shh response dynamics even after response initiation time re-alignment, distinct from the tracks from anterior datasets, suggesting an overall noisier environment in the posterior neural tube in zebrafish. This may be because posterior neural progenitors undergo more cell mixing, because their specification times are later or more variable, or because the spatially small Shh gradient as a result of the posterior neural tube's small size introduces more positional noise in signaling. Overall, simple response-fate models are insufficient to account for all the heterogeneities observed or to make precise fate predictions.

While it is tempting to construct more complex models using our single-cell data, certain limitations remain that warrant caution. First, our live imaging is limited by the brightness of reporters and the sensitivity of detection. Dynamics that fall below our imaging capacity, especially the Shh response activities before and in the early hours of our tracks, are not as well resolved, such that the initiation time of progenitor response and its heterogeneity may not be captured. Second, due to the constraint of a live embryo undergoing morphogenesis, certain cells cannot be fully tracked throughout the imaging time window. Third, even though most ventral cells are specified and no longer require Shh by the end of our tracks (17–18 hpf, *Huang et al., 2012*; *Xiong et al., 2013*), some cells may remain unspecified, and their fate reporter could further change beyond our detection. These limitations could give misleading results in overly fine-grained comparisons (e.g. response level within each 1 hr

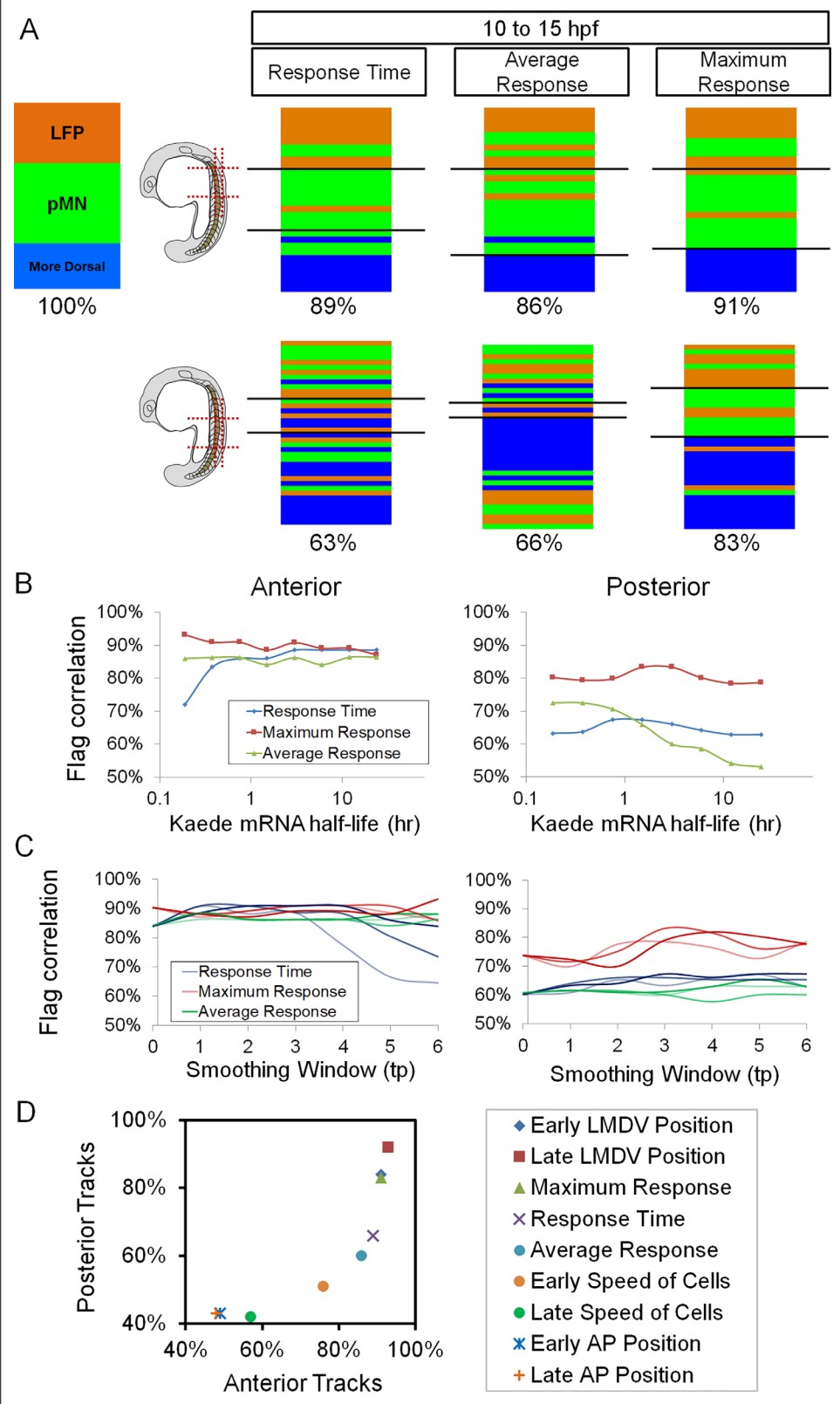

**Figure 7.** Correlations between different metrics of Sonic hedgehog (Shh) response and fate choice in single cells. (**A**) Shh response dynamics metrics. Under the modeled activity of the *ptch2:kaede* promoter, response time is a count of time points at which *ptch2:kaede* promoter is 'on' (>0.05 AU/hr²). Average response is the average promoter activity across the whole time window. Maximum response is the highest promoter activity found in the

*Figure 7 continued on next page*

*Figure 7 continued*

time window. (**B**) Flag correlation results of the three metrics in (**A**) (for which half-life=3 hr was used) over different Kaede mRNA half-life values. Note that response time and average response parameters switch predictive power as Kaede mRNA becomes stable. Maximum transient response level stays best over wide ranges of mRNA stability. The conclusion in (**A**) is therefore robust to this estimated parameter (*Figure 4*). (**C**) Flag correlation results of the three parameters in (**A**) (where smoothing window = 3 time points and iteration number = 2) over different smoothing windows (*x* axis) and iteration numbers (dark,1;light,2;lighter,3). The conclusion in (**A**) is therefore robust to this analysis, despite the sensitivity of *Equation 3* to the smoothing parameters. (**D**) Summary of correlation percentages for posterior vs anterior tracks for different morphogen interpretation models. In addition to the metrics mentioned in the main text, cell speeds were also plotted here. The speeds were calculated from the positions of the cells in the track for a 2 hr window centered on the early (11.5 hpf) and late (14.5 hpf) time points.

window, data not shown), where a much smaller number of tracks and time points could dominate and thus bias the analysis.

Nevertheless, the results presented here along with our previous study (*Xiong et al., 2013*) raise an important consideration for the understanding of morphogen interpretation: given the pervasive heterogeneity of single cells in position, timing, and signaling, how is patterning precision achieved? We speculate that precision at the single-cell level may not be necessary as the heterogeneity can average out on the population level, and thus important patterning features such as the size of each progenitor domain can remain robust even under single-cell noise. Still, it is important to note that the Shh GRN examined here already provides a strong degree of accuracy in single cells (e.g. Shh maximum response to fate choices) across datasets and explains a large part of the final pattern as seen on the population level. An additional morphogen gradient such as BMP from the dorsal neural tube could provide information in parallel with Shh to improve fate choice precision as a function of position (*Zagorski et al., 2017*), which is inherently challenging for a single morphogen gradient to convey (*Zagorski et al., 2024*). After cell fate choices were made, remaining positional errors can be corrected by cellular mechanisms such as cell sorting to refine the final pattern (*Tsai et al., 2020*). Finally, additional mechanisms such as apoptosis may provide means of correction. Further testing of these ideas would require improved imaging capacities and access to multiple nodes of the GRN with more accurate live reporters (e.g. fluorescent protein knock-ins). Larger-scale, more systematic single-cell imaging and analysis will allow a finer and potentially definitive quantitative dissection of this question.

## Materials and methods
### Zebrafish strains and maintenance
Zebrafish (*Danio rerio*) lines *Tg(shh:gfp)* (*Shkumatava et al., 2004*), *tgBAC(ptch2:kaede)* (*Huang et al., 2012*), *tg(nkx2.2a:mgfp)* (*Ng et al., 2005*), *tg(olig2:gfp)* (*Shin et al., 2003*), *tg(mnx1:gfp)* (*Flanagan-Steet et al., 2005*), and *tg(hsp70l:EGFP)* (*Halloran et al., 2000*) have been described. Homozygote parents from two different lines were paired to produce double transgenic embryos for imaging. Natural spawning was used, and time of fertilization was recorded at the single cell stage of each clutch. Embryos were kept in 28°C incubators/chambers during imaging and other times except room temperature during injections, dechorionating, and mounting. Staging was recorded using morphological criteria and aligned to the normal table (*Kimmel et al., 1995*). All fish are housed in fully equipped and regularly maintained and inspected aquarium facilities. Fish-related protocols have been approved by the Institutional Animal Care and Use Committee (IACUC) at Harvard Medical School under protocol # 04877.

### Microinjections of mRNAs
A pMTB construct (*Xiong et al., 2013*) containing *mem-EBFP2* or *h2b-EBFP2* was used for mRNA synthesis with the mMESSAGE mMACHINE system (Ambion). For mosaic injections, one blastomere of 8- to 16-cell stage embryos was injected (Nanoject) with approximately 1 nl of 20 ng/μl mem-EBFP2 or h2b-EBFP2. One round of screening was applied immediately after mosaic injections to eliminate damaged embryos and/or ones that missed the injection (no retention of co-injected Phenol Red label). Before imaging, injected embryos were screened for health.

## Timelapse confocal imaging

Live imaging was performed using a Zeiss 710 confocal microscope (objective: C-Apochromat 40× 1.2 NA) with a home-made heating chamber maintaining 28°C. Embryos were mounted using the dorsal mount (*Megason, 2009*) with a stereoscope (Leica MZ12.5). The mounting steps for imaging the neural plate/tube have been described in detail in *Xiong et al., 2013*. Laser was used for photo conversion of Kaede and excitation of EBFP2 at the same time. 488 nm and 561 nm were then used in a separate track for the GFP and Kaede (red) signals (see also *Figure 1B*). A typical stack covers an imaging space of 303 μm (*x*), 303 μm (*y*), and 120 μm (*z*), divided by 700×700×80 voxels and is taken within 6 min. A typical movie lasts ~10 hr containing ~100 stacks at 6 min intervals.

## Data analysis

### Track creation and fate assignment

For the present study, 31 timelapse datasets were collected (29 contain *ptch2:kaede* with *olig2:gfp*:15; *nkx2.2:mgfp*:6; *shh:gfp*:4; *mnx1:gfp*:4. 2 contain *olig2:dsRed* and *nkx2.2:mgfp*). 21/31 allow cell tracking and 13/21 have high coverage. Tracks were generated for 7/13 and 4/7 were further processed. The datasets acquired in Zeiss.lsm format were first converted to Megacapture format for import into GoFigure2 software (https://gofigure2.sourceforge.net/ see also *Xiong et al., 2013*; *Xiong et al., 2014*). Inside GoFigure2, segmentation and tracking were performed manually on each dataset on randomly selected ventral neural tube cells in the labeled embryo (e.g. *Figure 1D and E*). To calculate the cell's positions in relation to the embryo, notochord segmentations were generated every 10 time points (1 hr in real time) for each presented dataset. Segmentation and track tables exported from GoFigure2 were further processed using custom scripts (provided in accompanying supplemental data files) to generate the single slice labeled movies (*Video 2*). The movies were watched individually to determine the fate choice of the cell using a combinatorial criteria, including fate reporter expression and final position in the ventral neural tube, as described in *Xiong et al., 2013*. First, at 16 hpf, MFPs were identified as the centered, apically constricted, triangle-shaped column of cells immediately dorsal to the notochord (as illustrated in *Figure 1A*). This fate identification is checked with *shh:gfp* movies in which MFPs are GFP+. Next, LFPs are identified as the two columns of cells flanking both sides of the MFP (*Huang et al., 2012*). This fate identification is confirmed with *nkx2.2:mgfp* movies (e.g. *Video 1*). Next, pMNs are identified as the cells dorsal to the floor plate cells that are strongly *olig2:gfp*-positive. Finally, more dorsal cells are those that are further dorsal to the pMN domain and GFP- in *olig2:gfp* movies or that are located >40% ventral-dorsal distance at 16 hpf in movies without *olig2:gfp* signal. The segmentations, tracks, and cell fate information were then assembled through a preprocessing pipeline in MATLAB (MathWorks).

### Track processing

The output of the pipeline is a set of MATLAB matrices containing organized raw data for downstream analysis (e.g. *NewFinalWrkSpace_olig2_9.mat*, see supplemental data files). This organized raw data was then processed through a series of filters (provided in accompanying supplemental data files) to account for multiple experimental factors that introduce small errors in intensity measurements, including bleed-through, saturation, and noise, e.g., *SaturationMask.m*: a first filter that removes aberrations in intensity calculation arising from saturated pixels in the image. *BleedthroughCorr.m*: correcting bleed-through signal between two channels. Bleed-through correction runs automatically after the algorithm assesses the degree of bleed-through and then applies a percentage subtraction on the values to further reduce errors in intensity calculation. *SmoothData.m*: a smoothing filter that handles short-time fluctuations that come from segmentation variations and acquisition noise. Different utility scripts were coded to allow user interactions with both the raw and filtered data, as listed in AnalysisScript_List.xlsx.

### Track alignment

*ptch2:kaede* reporter tracks were aligned at the point at which the corresponding promoter activity traces crossed an activation threshold value and maintained it for three time points (18 min). The activation threshold was determined by comparing promoter activity values from more dorsally fated cells, in which the reporter is generally inactive, and those from LFP- and pMN-fated cells, in which

the reporter is activated. A threshold value of 0.025 could distinguish the two sets of traces, with only ~60% of more dorsally fated cells crossing the threshold at any point within the observation period compared to 100% of LFP- and pMN-fated cells.

## Track clustering analysis

ptch2:kaede reporter traces were clustered using the K-means clustering function in MATLAB. Each track was truncated to the set of time points for which >80% of reporter traces had observations. After truncation, tracks were only included in the analysis if they had observations for >80% of remaining time points. After filtering, 182 out of 216 traces were included in the clustering analysis. The elbow method was used to determine an appropriate number of clusters. The number of clusters was varied from 5 to 25, and for each value, the total within-cluster sum of squares (WCSS) value was calculated. For each cluster, the WCSS value corresponds to the sum of square distances for each point in the cluster from the centroid of the cluster. The total WCSS value decreases with an increasing number of clusters (because individual clusters become tighter) and begins to plateau at approximately 15 clusters. This value was therefore used for the analysis.

## Acknowledgements

We thank D D'India for fish care, K Mosaliganti, T Tsai, and members of the MATLAB community for sharing scripts, A Green, B Appel, and J Kuwada for transgenic lines, W Ma, T Mitchison, A Schier, and Megason lab members for discussions. This work is supported by NIH R01 GM107733 to SGTM and Natural Sciences and Engineering Research Council of Canada (NSERC) to PH. FX also acknowledges the support of a UKRI-EPSRC Frontier Research Grant (EP/X023761/1, originally selected as an ERC Starting Grant).

## Additional information

### Funding

| Funder | Grant reference number | Author |
|---|---|---|
| National Institutes of Health | GM107733 | Sean G Tsung-Megason |
| Engineering and Physical Sciences Research Council | EP/X023761/1 | Fengzhu Xiong |

The funders had no role in study design, data collection and interpretation, or the decision to submit the work for publication.

### Author contributions

Fengzhu Xiong, Conceptualization, Resources, Data curation, Software, Formal analysis, Validation, Investigation, Methodology, Writing – original draft, Writing – review and editing; Andrea R Tentner, Conceptualization, Resources, Data curation, Software, Formal analysis, Validation, Investigation, Methodology; Sandy Nandagopal, Data curation, Software, Formal analysis, Validation, Methodology; Tom W Hiscock, Resources, Data curation, Methodology; Peng Huang, Resources, Data curation, Validation, Investigation, Methodology; Sean G Tsung-Megason, Conceptualization, Supervision, Funding acquisition, Methodology, Writing – original draft, Project administration, Writing – review and editing

### Author ORCIDs

Fengzhu Xiong (iD) https://orcid.org/0000-0002-6153-0254
Andrea R Tentner (iD) https://orcid.org/0000-0002-0093-3367
Peng Huang (iD) https://orcid.org/0000-0001-7954-8869
Sean G Tsung-Megason (iD) https://orcid.org/0000-0002-9330-2934

### Ethics

Fish-related protocols have been approved by the Institutional Animal Care and Use Committee (IACUC) at Harvard Medical School under protocol # 04877.

Reviewer #1 (Public Review): https://doi.org/10.7554/eLife.96980.3.sa1
Reviewer #2 (Public Review): https://doi.org/10.7554/eLife.96980.3.sa2
Author response https://doi.org/10.7554/eLife.96980.3.sa3

## Additional files

### Supplementary files
MDAR checklist

### Data availability
Full analysis protocols, scripts and sample datasets are available for download here: https://github.com/xionglab/NT_heterogeneity (copy archived at *Xiong, 2024*).

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
