## [Editor Report · eLife Assessment]

This study presents an **important** study of the relationship between morphogen signaling and cell fate choices in the forming zebrafish neural tube, addressing a topical question in developmental biology. The authors provide a **solid** characterization of the precision limit for gene regulatory networks interpreting Shh, with single-cell resolution and state-of-the-art in vivo approaches. While the depth of analysis is restricted, particularly by the number of cell traces, the study will be of interest to developmental biologists interested in cellular decision-making.

---

## [Referee Report · Reviewer #1 (Public Review)]

[Editors' note: This version has been assessed by the Reviewing Editor without further input from the original reviewers. Given the time elapsed since the original data collection, the authors have addressed the previous concerns by providing a more nuanced discussion of their results and acknowledging the limitations of the study to ensure the conclusions are supported by the existing data.]

Throughout the paper, the authors do a fantastic job of highlighting caveats in their approach, from image acquisition to analysis. Despite this, some conclusions and viewpoints portrayed in this study do not appear well-supported by the provided data. Furthermore, there are a few technical points regarding the analysis that should be addressed.

(1) Analysis of signaling traces

- Relevance of "modeled signaling level": It is not clear whether this added complexity and potential for error (below) provides benefits over a more simple analysis such as taking the derivative (shown in Figure 3C). Could the authors provide evidence for the benefits? For example, does the "maximal response" given a simpler metric correlate less well with cell fate than that calculated from the fitted response?

- Assumptions for "modeled signaling level": According to equation (1) Kaede levels are monotonically increasing. This is assumed given the stability of the fluorescent protein. However, this only holds for the "totally produced Kaede/fluorescence". Other metrics such as mean fluorescence can very well decrease over time due to growth and division. Does "intensity" mean total fluorescence? Visual inspection of the traces shown in Figure 2 suggests that "fluorescence intensity" can decrease. What does this mean for the inferred traces?

- Estimation of Kaede reporter half-live: It is not clear how the mRNA stability of Kaede is estimated. It sounds like it was just assessed visually, which seems not entirely appropriate given the quantitative aspects of the rest of the study. Also, given that Shh signaling was inhibited on the level of Smoothened, it is not obvious how the dynamics of signaling shutdown affect the estimate. Most results in Figure 7 seem to be quite robust to the estimate of the half-live. That they are, might suggest that the whole analysis is unnecessary in the first place. However, not all are. Thus, it would be important to make this estimate more quantitative.

(2) Assignment of fates and correlations

- Error estimate for cell-type assignment: Trying to correlate signaling traces to cell fate decisions requires accurate cell fate assignment post-tracking. The provided protocol suggests a rather manual, expert-directed process of making those decisions. Can the authors provide any error-bound on those decisions, for example comparing the results obtained by two experts or something comparable? I am particularly concerned about the results regarding the higher degree of variability in the correlation between signaling dynamics and cell fate in the posterior neural tube. Here, the expression of Olig2 does not seem to segregate between different assigned fates, while it does so nicely in the anterior neural tube. This would suggest to me that cells in the posterior neural tube might not yet be fully committed to a fate or that there could be a relatively high error rate in assigning fates. Thus, the results could emerge from technical errors or differences in pure timing. Could the authors please comment on these possibilities?

- Clustering and fates: One approach the authors use to analyze the correlation between signaling and fate is clustering of cell traces and comparison of the fate distributions in those clusters. There is a large number of clusters with only single traces, suggesting that the data (number of traces) might not be sufficient for this analysis. Furthermore, I am skeptical about clustering cells of different anterior-posterior identities together, given potential differences in the timing of signal reception and signaling. I am not convinced that this analysis reveals enough about how signaling maps to fate given the heterogeneity in traces in large clusters and the prevalence of extremely small clusters.

- Signaling vector and hand-picked metrics: As an alternative approach, that might be better suited for their data, the authors then pick three metrics (based on their model-predicted signaling dynamics) and show that the maximal response is a very good predictor of fate for different anterior-posterior identities. Previous information-theoretic analysis of signaling dynamics has found that a whole time-vector of signaling can carry much more information than individual metrics (Selimkhanov et al, 2014, PMID: 25504722). Have the authors tried to use approaches that make use of the whole trace such as simple classifiers (Granados et al, 2018, PMID: 29784812), or can comment on why this is not feasible for their data? The authors should at least make clear that their results present a lower bound to how accurately cells can make cell-fate decisions based on signaling dynamics.

(3) Consequences of signaling heterogeneity

The authors focus heavily on portraying that signaling dynamics are highly variable, which seems visually true at first glance. However, there is no metric used or a description given of what this actually means. Mainly, the variability seems to relate to the correlation between signaling and fate. However, given the data and analysis, I would argue that the decoding of signaling dynamics into fate is surprisingly accurate. So signaling dynamics that seem quite noisy and variable by visual inspection can actually be very well discriminated by cells, which to me appears very exciting.

Indeed, simple features of signaling traces can predict cell fate as well as position (for anterior progenitors). Given that signaling should be a function of position, it naively seems as if signaling read-out could be almost perfect. It might be interesting to plot dorsal-ventral position vs the signaling metrics, to also investigate how Shh concentration/position maps to signaling dynamics, this would give an even more comprehensive view of signal transmission.

There remains the discrepancy between signaling traces and fate in the posterior neural tube. The authors point towards differences in tissue architecture and difficulties in interpreting a "small" Shh gradient. However, the data seems consistent with differences in timing of cell-fate decisions between anterior and posterior cells. The authors show that fate does initially not correlate well with position in the posterior neural tube. So, signaling dynamics should likely also not, as they should rather be a function of position, given they are downstream of the Shh gradient. As mentioned above, not even Olig2 expression does segregate the assigned fates well. All this points towards a difference in the time of fate assignment between the anterior and posterior. Given likely delays in reporter protein production and maturation, it can thus not be expected that signaling dynamics correlate better with cell fate than the reporter "83%". Can the authors please discuss this possibility in the paper?

Thus, while this paper represents an example of what the community needs to do to gain a better understanding of robust patterning under variability, the provided data is not always sufficient to make clear conclusions regarding the functional consequences of signaling dynamics.

---

## [Referee Report · Reviewer #2 (Public Review)]

Summary:

In this work, Xiong and colleagues examine the relationship between the profile of the morphogen Shh and the resulting cell fate decisions in the zebrafish neural tube. For this, the authors combine high-resolution live imaging of an established Shh reporter with reporter lines for the different progenitor types arising in the forming neural tube. One of the key observations in this manuscript is that, while, on average, cells respond to differences in Shh activity to adopt distinct progenitor fates, at the single cell level there is strong heterogeneity between Shh response and fate choices. Further, the authors showed that this heterogeneity was particularly prominent for the pMN fate, with similar Shh response dynamics to those observed in neighboring LFP progenitors.

Strengths:

It is important to directly correlate Shh activity with the downstream TFs marking distinct progenitor types in vivo and with single cell resolution. This additional analysis is in line with previous observations from these authors, namely in Xiong, 2013. Further, the authors show that cells in different anterior-posterior positions within the neural tube show distinct levels of heterogeneity in their response to Shh, which is a very interesting observation and merits further investigation.

Weaknesses:

This is a convincing work, however, adding a few more analyses and clarifications would, in my view, strengthen the key finding of heterogeneity between Shh response and the resulting cell fate choices.

---

## [Author Response]

The following is the authors’ response to the original reviews.

**Public Reviews:**

**Reviewer #1 (Public Review):**
Throughout the paper, the authors do a fantastic job of highlighting caveats in their approach, from image acquisition to analysis. Despite this, some conclusions and viewpoints portrayed in this study do not appear well-supported by the provided data. Furthermore, there are a few technical points regarding the analysis that should be addressed.

We thank the reviewer for the comments, due to the age of the work and logistic constraints, we are unable to perform further experiments and analysis to address some of the concerns. We revised conclusions and viewpoints accordingly to reflect reviewer concerns.

(1) Analysis of signaling tracesRelevance of "modeled signaling level": It is not clear whether this added complexity and potential for error (below) provides benefits over a more simple analysis such as taking the derivative (shown in Figure 3C). Could the authors provide evidence for the benefits? For example, does the "maximal response" given a simpler metric correlate less well with cell fate than that calculated from the fitted response?

We think the benefits of modeled signaling level are the conceptual accuracy to the extent possible with the data. It’s true that the assumptions brought-in may cause certain biases. We perform this and the simplest (raw data averaging, Fig.2). Intermediate results in between (such as the first derivative in Fig.3C) may correlate well or less well, but cannot be interpreted biologically.

Assumptions for "modeled signaling level": According to equation (1) Kaede levels are monotonically increasing. This is assumed given the stability of the fluorescent protein. However, this only holds for the "totally produced Kaede/fluorescence." Other metrics such as mean fluorescence can very well decrease over time due to growth and division. Does "intensity" mean total fluorescence? Visual inspection of the traces shown in Figure 2 suggests that "fluorescence intensity" can decrease. What does this mean for the inferred traces?

Yes the segmentations measure intensity in a fixed volume inside a cell, therefore it’s a spatial average (concentration) and is susceptible to cell volume changes. This has been noted in the revision. The raw measurement does fluctuate and can decrease, we think the short-time-scale fluctuations are likely measurement variations/errors rather than underlying big changes in concentration.

Estimation of Kaede reporter half-live: It is not clear how the mRNA stability of Kaede is estimated. It sounds like it was just assessed visually, which seems not entirely appropriate given the quantitative aspects of the rest of the study. Also, given that Shh signaling was inhibited on the level of Smoothened, it is not obvious how the dynamics of signaling shutdown affect the estimate. Most results in Figure 7 seem to be quite robust to the estimate of the half-live. That they are, might suggest that the whole analysis is unnecessary in the first place. However, not all are. Thus, it would be important to make this estimate more quantitative.

Yes we agree. Unfortunately we don’t have the quantitative data required to better estimate Kaede mRNA stability. The timing of Cyc inhibition to the ceasing of ptch mRNA production is roughly estimated but not necessarily precise in this context.

(2) Assignment of fates and correlationsError estimate for cell-type assignment: Trying to correlate signaling traces to cell fate decisions requires accurate cell fate assignment post-tracking. The provided protocol suggests a rather manual, expert-directed process of making those decisions. Can the authors provide any error-bound on those decisions, for example comparing the results obtained by two experts or something comparable? I am particularly concerned about the results regarding the higher degree of variability in the correlation between signaling dynamics and cell fate in the posterior neural tube. Here, the expression of Olig2 does not seem to segregate between different assigned fates, while it does so nicely in the anterior neural tube. This would suggest to me that cells in the posterior neural tube might not yet be fully committed to a fate or that there could be a relatively high error rate in assigning fates. Thus, the results could emerge from technical errors or differences in pure timing. Could the authors please comment on these possibilities?

This is a very insightful point. We did examine the posterior data again (cross-checked by 2 co-authors) to make sure the mixed situation has correct cell fate assignment. As established by others’ and our previous studies (See also Fig.1A), the identification of MFPs and LFPs in zebrafish spinal cord is very robust. The MFPs are the apical constricted single column of cells along the midline on top of the notochord, and the LFPs are the 2 columns of cells next to MFP on both sides. LFPs’ expression of olig2:gfp did vary more in the posterior (timing of response/commitment could be a factor as the reviewer pointed out), but eventually the cells at those positions will be V3 interneurons or floor plates and have not been observed to make motoneurons. There are 3 low Olig2:GFP pMNs in the anterior dataset (Fig.2B’) and 3 high Olig2:GFP LFPs in the posterior dataset (Fig.2D’) that we checked carefully. The heterogeneity argument is based on the verified tracking and final positioning of these cells.

Clustering and fates: One approach the authors use to analyze the correlation between signaling and fate is clustering of cell traces and comparison of the fate distributions in those clusters. There is a large number of clusters with only single traces, suggesting that the data (number of traces) might not be sufficient for this analysis. Furthermore, I am skeptical about clustering cells of different anterior-posterior identities together, given potential differences in the timing of signal reception and signaling. I am not convinced that this analysis reveals enough about how signaling maps to fate given the heterogeneity in traces in large clusters and the prevalence of extremely small clusters.

We agree. Due to the age of the work and logistic constraints, we are unable to perform further experiments and analysis to enrich the tracks for this revision. We are aware of upcoming, independent studies with many more systematic tracks and analysis which will address these concerns. We have added the caveats the reviewer raised.

Signaling vector and hand-picked metrics: As an alternative approach, that might be better suited for their data, the authors then pick three metrics (based on their model-predicted signaling dynamics) and show that the maximal response is a very good predictor of fate for different anterior-posterior identities. Previous information-theoretic analysis of signaling dynamics has found that a whole time-vector of signaling can carry much more information than individual metrics (Selimkhanov et al, 2014, PMID: 25504722). Have the authors tried to use approaches that make use of the whole trace such as simple classifiers (Granados et al, 2018, PMID: 29784812), or can comment on why this is not feasible for their data? The authors should at least make clear that their results present a lower bound to how accurately cells can make cell-fate decisions based on signaling dynamics.

Thanks for these suggestions. We are limited by the measurement noise, coverage window of the traces and the number of tracks to make use of the full dynamics in a more informative manner.

(3) Consequences of signaling heterogeneityThe authors focus heavily on portraying that signaling dynamics are highly variable, which seems visually true at first glance. However, there is no metric used or a description given of what this actually means. Mainly, the variability seems to relate to the correlation between signaling and fate. However, given the data and analysis, I would argue that the decoding of signaling dynamics into fate is surprisingly accurate. So signaling dynamics that seem quite noisy and variable by visual inspection can actually be very well discriminated by cells, which to me appears very exciting.

Yes – we agree that most cells are actually accurate in such a highly dynamic tissue. In the literature, the view has been more focused on how the GRN enables this accuracy. We therefore highlighted the heterogeneity and limit of accuracy of the GRN here. We added this point to make our presentation more balanced.

Indeed, simple features of signaling traces can predict cell fate as well as position (for anterior progenitors). Given that signaling should be a function of position, it naively seems as if signaling read-out could be almost perfect. It might be interesting to plot dorsal-ventral position vs the signaling metrics, to also investigate how Shh concentration/position maps to signaling dynamics, this would give an even more comprehensive view of signal transmission.

We’d refer readers to our earlier study Xiong et al., 2013 where ptch2:kaede, nkx2:gfp and olig2:gfp were plotted against position over time in single cell tracks. It was found that position was not a good predictor of signaling levels or cell fates at early stages when the cell fates were specified.

There remains the discrepancy between signaling traces and fate in the posterior neural tube. The authors point towards differences in tissue architecture and difficulties in interpreting a "small" Shh gradient. However, the data seems consistent with differences in timing of cell-fate decisions between anterior and posterior cells. The authors show that fate does initially not correlate well with position in the posterior neural tube. So, signaling dynamics should likely also not, as they should rather be a function of position, given they are downstream of the Shh gradient. As mentioned above, not even Olig2 expression does segregate the assigned fates well. All this points towards a difference in the time of fate assignment between the anterior and posterior. Given likely delays in reporter protein production and maturation, it can thus not be expected that signaling dynamics correlate better with cell fate than the reporter "83%". Can the authors please discuss this possibility in the paper?

Yes this is an important point/caveat of live signaling and fate tracking. As discussed in the manuscript, due to the sensitivity limit of fluorescent imaging, it’s difficult to determine the time when cells start to respond to the signal, and how variable that is from cell to cell. The posterior cells may be more variable in either spatial or temporal responses compared to the anterior and we are not able to distinguish that. However, signaling dynamics is not necessarily a good function of position or time either, there is no evidence for that in our results here. The 83% correlation is thus striking for the posterior progenitors indicating a certain robust logic in the GRN to capture a strong (even short-lived) response to Shh, regardless of position or time. This is an interest possibility (we do not claim it a mechanism as we have not tested it with perturbations) that challenges the prevailing view in the field that these progenitors integrate Shh exposure over time, or that they acquire positional information by reading a gradient.

The discussion has been modified to be more nuanced about these points.

Thus, while this paper represents an example of what the community needs to do to gain a better understanding of robust patterning under variability, the provided data is not always sufficient to make clear conclusions regarding the functional consequences of signaling dynamics.

We quite agree. Together with the reviewer, we look forward to seeing the publication of some recent, independent progresses overcoming the challenges in our work by other colleagues.

**Reviewer #2 (Public Review):**
Summary:In this work, Xiong and colleagues examine the relationship between the profile of the morphogen Shh and the resulting cell fate decisions in the zebrafish neural tube. For this, the authors combine high-resolution live imaging of an established Shh reporter with reporter lines for the different progenitor types arising in the forming neural tube. One of the key observations in this manuscript is that, while, on average, cells respond to differences in Shh activity to adopt distinct progenitor fates, at the single cell level there is strong heterogeneity between Shh response and fate choices. Further, the authors showed that this heterogeneity was particularly prominent for the pMN fate, with similar Shh response dynamics to those observed in neighboring LFP progenitors.Strengths:It is important to directly correlate Shh activity with the downstream TFs marking distinct progenitor types in vivo and with single cell resolution. This additional analysis is in line with previous observations from these authors, namely in Xiong, 2013. Further, the authors show that cells in different anterior-posterior positions within the neural tube show distinct levels of heterogeneity in their response to Shh, which is a very interesting observation and merits further investigation.Weaknesses:This is a convincing work, however, adding a few more analyses and clarifications would, in my view, strengthen the key finding of heterogeneity between Shh response and the resulting cell fate choices.

We thank the reviewer for the comments, due to the age of the work and logistic constraints, we are unable to perform further experiments and analysis to address some of the concerns. We revised conclusions and viewpoints accordingly to reflect reviewer concerns.

**Recommendations for the authors:**

**Reviewer #1 (Recommendations for The Authors):**
Minor comments:y-axis label suddenly changes to Ptch2-reporter level in Figure 5. Is what is plotted different from what is seen as examples in Figure 3?

Thanks! Figure 5 tracks are as Figure 3B, this has been annotated in the figure legends.

There are random bounding boxes in some of the figures.Sometimes the m in "More dorsal" is stylized with a capital M and sometimes not. It is somewhat confusing as a name for cell types but it is fine if no alternative can be found.

This study unfortunately does not include markers that distinguish the interneurons dorsal to pMNs. We categorized them collectively as “more dorsal”.

Response-time is defined as "the amount of time with an above-basal Shh response". This seems to me as the definition of response duration. I would assume that response-time, means the time it takes until a response is first observed. Please consider changing this.

We did not use “duration” because a response time course recorded in these tracks may include multiple durations (on and off). The duration of exposure/response has been specifically used in the field as a single period of response. So it’s a sum of active responding time here. Clarified in the text.

**Reviewer #2 (Recommendations for The Authors):**
(1) The authors address several possible setbacks of transforming the measured fluorescence intensity of the patched reporter into a readout of the Shh signaling activity over time, however, one aspect that isn't directly addressed is the potential effect of differences in the z position of analyzed cells. These could, at least in principle, be sufficient to introduce significant noise in the fluorescence measurements. Can the authors subset their datasets by initial, as well as average, z position and then re-examine the measured trends for both Shh activity and the intensity of the cell fate reporters used in the study?

The zebrafish early neural plate/tube has a small thickness in z in dorsal-ventral imaging and the tissue is transparent. The depth-associated scattering contributes very little, if at all to the fluorescent signals in the imaged time window. This can be seen in the nuclear/membrane signal of the movies, which is largely uniform across the tissue in z in the neural tissue. It can also be seen that the notochord cells, further ventral, appears to be dimmer.

(2) It is critical for the validity of this study that the intensity of the patched reporter introduced by the authors in 2012, and used again in this study, faithfully represents the signaling activity of Shh. In this study, the authors provide measurements of the transcriptional rate of Kaede and additional modeling for this purpose. However, an important point is to determine how sensitive is the reporter to changes in Shh signaling of different magnitudes?

We consider this BAC reporter line a good (probably still the best live reporter) one as it resolves the endogenous gradient up to the dorsal interneuron domains (Huang et al., 2012, Xiong et al., 2013) and responds well to perturbations (Notch, Cyclopamine, etc). But it’s true that we don’t have information of how sensitive it responds to changes of different magnitude. As far as we know, there is no in vivo, single cell information of how Shh targets respond to signaling of different magnitudes.

(3) To strengthen the previous point, it would be nice to extend the analysis in Figure 2, at least partially, using other readouts for Shh activity (e.g. GBS-GFP)?

We have used a GBS-RFP line previously and found it to be lower resolution in terms of showing the DV gradient, compared to ptch2:kaede.

(4) It is unclear to me what is the relevant time window during which cells respond to Shh in the anterior versus posterior domains to determine progenitor specification. This is a concern to me, since: (i) the average heterogeneity of Shh activity seems to increase strongly in time (Figure 2A/C); and (ii) it is important to exclude that the finding of heterogeneous relationship between Shh activity and fate choices is largely driven by later timepoints, where potentially its activity is no longer relevant for cell fate specification. Can this point be clarified when this data is introduced in the manuscript and further discussed?

Yes this is an important point/caveat of live signaling and fate tracking. As discussed in the manuscript, due to the sensitivity limit of fluorescent imaging, it’s difficult to determine the time when cells start to respond to the signal, and how variable that is from cell to cell. The posterior cells may be more variable in either spatial or temporal responses compared to the anterior and we are not able to distinguish that.

(i) The ptch2:kaede reporter variability is higher in terms of magnitude (the signal gets brighter) in later times but the heterogeneity (overlap between difference cell fate groups) is lower in later times

(ii) Similarly, the heterogenous relationship is more pronounced in early time points. Since we do not know exactly when the activity becomes no longer relevant (from our earlier studies we do think that the cells become specified early, when Shh signaling is noisy), we modelled the response profile and searched for a good predictor. The maximum response stands out, particularly as a good indicator for the posterior cells, suggests an early window/time of specification.

Discussion has been modified to clarify these points.

(5) Is the response of the patched reporter, as well as cell fate reporters, to defined concentrations of exogenously provided Shh heterogeneous, for instance, in in vitro experiments?

Well-controlled (e.g., microfluidics and labeled Shh molecules) in vitro experiments will be fantastic future directions. Existing tissue explant + Shh dose approaches do not resolve the heterogeneity of exposure at single cell level but may be helpful in testing the limits and variabilities at different magnitudes.

(6) The source of noise in this system is not entirely clear to me. The authors seem to attribute the heterogeneity they observe to the way cells respond to Shh, but can it be excluded that the morphogen profile is itself noisy to start with? It is currently difficult to distinguish between these two possibilities, given that the Shh activity reporter used in this study is itself a transcriptional output of the pathway. Can the distribution of Shh itself be analyzed (even if in immunostainings) during neural tube formation?

Yes we fully agree. More quantitative analysis may help dissecting the sources of noise. The morphogen profile (particularly through time) will be great. Currently no reagent is available to achieve that. Studies using an engineered morphogen or tagged morphogen suggest that the pattern through tissue reasonably captures simple diffusion dynamics. However, at single cell level considerable randomness may still remain and difficult to quantitatively compare with still staining.

(7) It is unclear to me how the authors define the ultimate cell fate of cells in their analysis in Figure 6. The brief description in the methods and in the manuscript seems to suggest that, in combination with marker expression, the cell position is used as a criteria to assign the fate to the progenitors - if this is the case, I guess the observed relationship in Figure 6 with LMDV distance is almost a control? This could be clarified for the readers.

Yes indeed Figure 6 is a control as LMDV distances lead to final positions which form part of our determination of cell fates.

As established by others’ and our previous studies (See also Fig.1A), the identification of MFPs and LFPs in zebrafish spinal cord is very robust. The MFPs are the apical constricted single column of cells along the midline on top of the notochord, and the LFPs are the 2 columns of cells next to MFP on both sides. LFPs’ expression of olig2:gfp did vary more in the posterior (timing of response/commitment could be a factor as the reviewer pointed out), but eventually the cells at those positions will be V3 interneurons or floor plates and have not been observed to make motoneurons. There are 3 low Olig2:GFP pMNs in the anterior dataset (Fig.2B’) and 3 high Olig2:GFP LFPs in the posterior dataset (Fig.2D’) that we checked carefully.

The methods of fate determination are described in detail in methods.

(8) The graphs in Figures 6 and 7 are difficult to interpret. What proportion, and absolute number, of cells are "mis specified" when the authors show the distinct colored lines in the pMN, LFP or more dorsal domains? How do the authors determine where each cell fate domain begins and ends to access for "mis-specified" cells? Can the authors also provide the corresponding experimental images in the figure?

We apologize for the difficulties to interpret these figures. The graphs are a ranked list of all cells using the specified metric. The visual is to help generate an intuition of how mixed vs clear-cut the pattern is given the tested metric. They are not to be interpreted as the actual pattern in the tissue and there are no data images that show these patterns.

(9) Given the experimental limitations/technical challenges discussed by the authors during the paper, the score of around 90% of predictability of cell fate choices is rather high in the anterior domain, suggesting a minor functional role for heterogeneity in this region. Even for the posterior domain, the score of 83% predictability based on the maximum response to Shh is still relatively high. In my view, this author's conclusions should be adjusted to make this difference clearer in the abstract and discussion, highlighting that the heterogeneity between Shh response and cell fate choices, particularly in the pMN fate, are stronger in the posterior domain affecting the precision of cell fate decisions particularly in this region. Can the authors further comment on potential mechanisms driving this difference?

Yes – we agree that most cells are actually accurate in such a highly dynamic tissue. In the literature, the view has been more focused on how the GRN enables this accuracy. We therefore highlighted the heterogeneity and limit of accuracy of the GRN here.

We have added the fact that the Shh response is still the main determinant of the pattern despite the heterogeneity in the Discussion. We also further discussed possibilities of the anterior posterior differences.

(10) Following up from the previous point, the data in Figure 7 suggests that there might be different underlying mechanisms in how anterior and posterior cells interpret the Shh profile, with anterior cells potentially responding to the integrated concentration of Shh (since response time, average response, or maximum response to Shh all provide similar predictability scores for cell fate choices). In contrast, only the maximum response to Shh can provide a good prediction of posterior cell fate, consistent with a more instantaneous response to morphogen concentration (and thus potentially more error-prone measurement of the Shh profile?). This is a very interesting observation in my view. Could this be further tested?

Thank you. Yes we found this very interesting too. We discussed the possibilities, including the reviewer’s suggestion that these cells may have different contexts or strategy to interpret the signal. It is also possible that the anterior cells use the same strategy (maximum response at an early time) and the subsequent response/duration do not matter to their fate commitment. A precise approach to shut down Shh response dynamics in single cells (e.g., optogenetics) will enable the test of these ideas. We hope following up studies will take such approaches.